# Connective Cognition Network for Directional Visual Commonsense Reasoning

**Aming Wu**[1][*]    **Linchao Zhu**[2]    **Yahong Han**[1][†]    **Yi Yang**[2]

[1]College of Intelligence and Computing, Tianjin University, Tianjin, China
[2]ReLER, University of Technology Sydney, Australia
{tjwam,yahong}@tju.edu.cn, {Linchao.Zhu, yi.yang}@uts.edu.au

## Abstract

Visual commonsense reasoning (VCR) has been introduced to boost research of cognition-level visual understanding, i.e., a thorough understanding of correlated details of the scene plus an inference with related commonsense knowledge. Recent studies on neuroscience have suggested that brain function or cognition can be described as a global and dynamic integration of local neuronal connectivity, which is context-sensitive to specific cognition tasks. Inspired by this idea, towards VCR, we propose a connective cognition network (CCN) to dynamically reorganize the visual neuron connectivity that is contextualized by the meaning of questions and answers. Concretely, we first develop visual neuron connectivity to fully model correlations of visual content. Then, a contextualization process is introduced to fuse the sentence representation with that of visual neurons. Finally, based on the output of contextualized connectivity, we propose directional connectivity to infer answers or rationales. Experimental results on the VCR dataset demonstrate the effectiveness of our method. Particularly, in $Q \to AR$ mode, our method is around 4% higher than the state-of-the-art method.

## 1 Introduction

Recent advances in visual understanding mainly make progress on the recognition-level perception of visual content, e.g., object detection [13, 23] and segmentation [9, 5], or even on the recognition-level grounding of visual concepts with image regions, e.g., image captioning [40, 24] and visual question answering [1, 6]. Towards complete visual understanding, a model must move forward from perception to reasoning, which includes cognitive inferences with correlated details of the scene and related commonsense knowledge. As a key step towards complete visual understanding, the task of Visual Commonsense Reasoning (VCR) [42] is proposed along with a well-devised new dataset. In VCR, given an image, a machine is required to not only answer a question about the thorough understanding of the correlated details of the visual content, but also provide a rationale, e.g., contextualized with related visual details and background knowledge, to justify why the answer is true. As a first attempt to narrow the gap between recognition- and cognition-level visual understanding, Recognition-to-Cognition Networks (R2C) [42] conducts visual commonsense reasoning step by step, i.e., grounding the meaning of natural language with respect to the referred objects, contextualizing the meaning of an answer with respect to the question and related global objects, and finally reasoning over the shared representation to obtain a decision of an answer. Due to the large discrepancy between the reasoning scheme of VCR and cognition function of human brain, R2C's performance is not in competition with humans score, e.g., 65% vs. 91% in $Q \to A$ mode.

---

[*]This work was done when Aming Wu visited ReLER Lab, UTS.
[†]Corresponding author

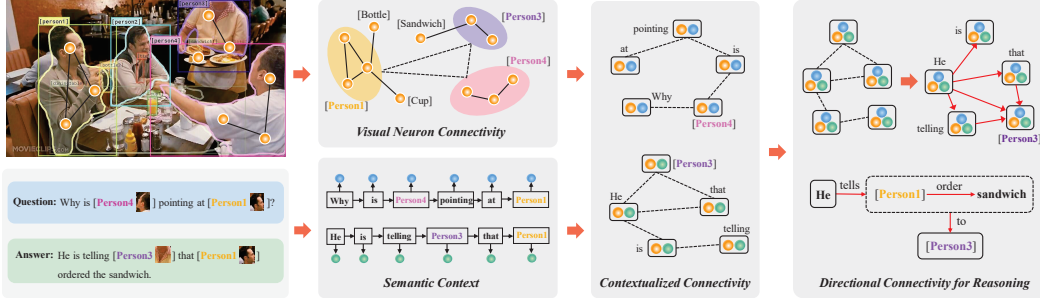

Figure 1: Overview of our CCN method. The yellow, blue, and green circles indicate visual elements, question and answer representation, respectively. Our method mainly includes visual neuron connectivity, contextualized connectivity, and directional connectivity. For semantic context, two LSTM units are used to extract sentence representations.

Recent studies [31, 8] on brain networks have suggested that brain function or cognition can be described as the global and dynamic integration of local (segregated) neuronal connectivity. And such a global and dynamic integration is context-sensitive with respect to a specific cognition task. Inspired by this idea, in this paper, we propose a Connective Cognition Network (CCN) for visual commonsense reasoning. As is shown in Fig. 1, the main process of CCN is to dynamically reorganize (integrate) the visual neuron connectivity that is contextualized by the meaning of answers and questions in the current reasoning task.

Concretely, taking visual words as visual neurons and object features as segregated visual modules, we first devise an approach of Conditional GraphVLAD to represent image's visual neuron connectivity, which includes connections among visual neurons and visual modules. The visual neuron connectivity serves as the base function for the dynamic integration in the process of reasoning. Meanwhile, as a context-sensitive integration, the meaning is specified by the semantic context of questions and answers. After obtaining the sequential information of sentences via an LSTM network [16], we fuse the sentence representation with that of the visual neurons, which stands for a contextualization.

Then we employ graph convolution neural network (GCN) to fully integrate both the local and global connectivity. For example, in Fig. 1, connections between "He" and "Person4", "Person4" and "Person3", as well as "Person3" and "table" could all be incorporated in the contextualized connectivity, where the last connection between "Person3" and "table" belongs to the global integration not mentioned here. Though the contextualized connectivity is ready for reasoning, it lacks direction information, which is an important clue for cognitive reasoning [32]. Taking the answer sentence in Fig. 1 as an example, there exists directional connection from "Person4" to "Person3" via the predicate "tell", as well as from "Person1" to "sandwich" via the predicate "order". Though easy to be defined in first-order logic (FOL) [36], it is nontrivial to be incorporated into a data-driven learning process. In this paper, we make an attempt to devise a direction learner on the GCN, so as to further improve the reasoning performance. Particularly, a network is first used to learn the semantic direction of input features. Then, we add the direction to the computation of the adjacency matrix of GCN to obtain a directional adjacency matrix, which serves as directional connectivity for reasoning.

Thus, we develop a novel connective cognition network for directional visual commonsense reasoning. The main contributions lie in that, this is the first attempt to use an end-to-end training neural network for the cognitive reasoning process, i.e., global and dynamic integration of local (segregated) visual neuron connectivity, which is context-sensitive with respect to a specific VQA task. Moreover, we also try to incorporate directional reasoning into a data-driven learning process. Experimental results on the VCR dataset [42] demonstrate the effectiveness of the proposed method. On the three reasoning modes of VCR task, i.e., $Q \rightarrow A$, $QA \rightarrow R$, and $Q \rightarrow AR$, the CCN with directional reasoning significantly outperforms R2C by 3.4%, 3.2%, and 4.4%, respectively.

## 2    Related Work

**Visual Question Answering:** Recently, many effective methods are proposed in the VQA task, which includes those based on attention [21, 26], multi-modal fusion [33, 12], and visual reasoning

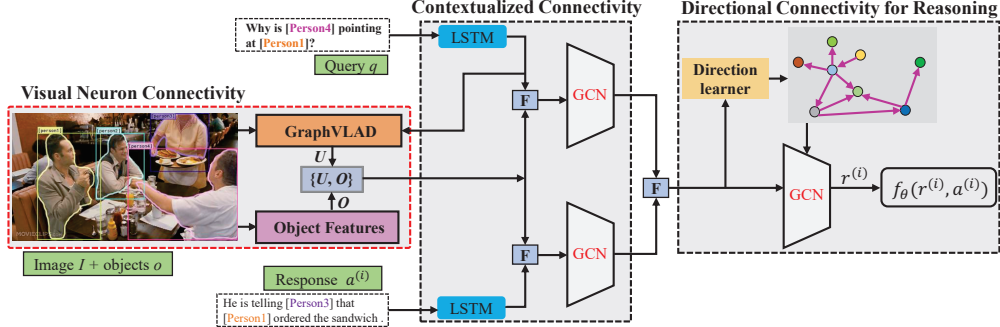

Figure 2: The framework of the CCN method. It mainly includes visual neuron connectivity, contextualized connectivity, and directional connectivity for reasoning. Here, '$\{U, O\}$' indicates the set including the output $U$ of GraphVLAD and object features $O$. '$f_\theta$' indicates the prediction function for responses (answers or rationales). 'F' indicates fusion operation.

[35, 29]. Most methods focus on the recognition of visual content and spatial positions, but they lack the ability of commonsense reasoning. To advance the research of reasoning, a new task of VCR [42] is proposed. Given a query-image pair, this task needs models to choose correct answer and rationale justifying why the answer is true. The challenges mainly include a thorough understanding of vision and language as well as a method to infer responses (answers or rationales). In this paper, we propose a CCN model for VCR, which has been proved to be effective in the experiment.

**NetVLAD:** The work [4] proposes NetVLAD which is used to extract local features. Particularly, it includes an aggregation layer for clustering the local features into a VLAD [19] global descriptor. Recently, NetVLAD has been demonstrated to be effective in many tasks [2, 37]. Particularly, the work [2] proposes a PointNetVLAD to extract the global descriptor from a given 3D point cloud. Besides, the state-of-the-art models [7, 27] of video classification most use NetVLAD pooling to aggregate information from all the frames of a video. However, the original NetVLAD learns multiple centers from the overall dataset to represent each input data, which ignores the characteristic of the input data and reduces the accuracy of the representation. To alleviate this problem, in this paper, we propose a conditional GraphVLAD to integrate the characteristic of the input data.

**Graph Convolutional Network:** GCN [22, 39, 28, 43] aims to generalize the Convolutional Neural Network (CNN) to graph-structured data. By encoding both the structure of the graph surrounding a node and the feature of the node, GCN could learn representation for every node effectively. As GCN has the benefit of capturing relations between nodes, many works have employed GCN for reasoning [17, 29]. Particularly, the work [29] uses GCN to infer answers. However, it only constructs an undirected graph for reasoning [29], which ignores the directional information between nodes. The directional information is often considered an important factor for inference [32]. Here, we propose a directional connectivity to infer answers, which has been proved to be effective.

## 3 Connective Cognition Network

Fig. 2 shows the framework of CCN model. It mainly includes visual neuron connectivity, contextualized connectivity, and directional connectivity for reasoning.

### 3.1 Visual Neuron Connectivity

The goal of visual neuron connectivity (Fig. 3(a)) is to obtain a global representation of an image, which is helpful for a thorough understanding of visual content. It mainly includes visual element connectivity and the computation of both conditional centers and GraphVLAD.

**Visual Element Connectivity.** We first use a pre-trained network, e.g., ResNet [15], to obtain the feature map $X \in \mathbb{R}^{w \times h \times m}$ of an image, where $w$, $h$, and $m$ separately indicate the width, height, and number of channels. Here, we take each element of the feature map as a visual element. We take the output $Y \in \mathbb{R}^n$ of LSTM [16] at the last time step as the representation of query (question or question with a correct answer).

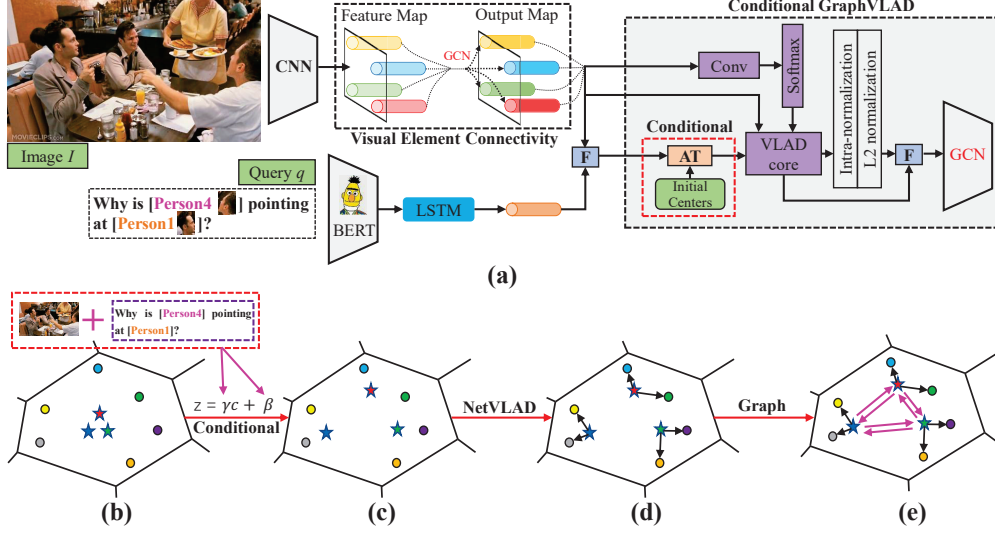

**(a)**

**(b)**         **(c)**         **(d)**         **(e)**

Figure 3: (a) shows the process of visual neuron connectivity. 'AT' indicates affine transformation. (b) shows the initial state of NetVLAD. (c) shows the conditional centers after an affine transformation. Here, we use the fusion of image and question to compute the parameter $\gamma$ and $\beta$. (d) and (e) show the result of NetVLAD and GraphVLAD, respectively.

In general, there exists certain relation between objects of an image [10]. As is shown in the left part of Fig. 1, relations (solid and dotted lines) exist not only between elements (yellow circles) in the same object region, but also between various objects (Person1, Person3, Person4, and background). Obviously, capturing these relations is helpful for a thorough understanding of the entire scene. In this paper, we employ GCN to capture these relations. Specifically, we seek to construct an undirected graph $G_g = \{V, \xi, \mathbf{A}\}$, where $\xi$ is the set of graph edges to learn and $\mathbf{A} \in \mathbb{R}^{N \times N}$ ($N = wh$) is the corresponding adjacency matrix. Each node $\nu \in V$ corresponds to one element of the feature map. And the size of $V$ is set to $N$. We first reshape $X$ to $\widetilde{X} \in \mathbb{R}^{N \times m}$. Then, we define an adjacency matrix for an undirected graph as $\mathbf{A} = softmax_r(\widetilde{X}\widetilde{X}^T) + I_d$, where $I_d$ indicates the identity matrix and $softmax_r$ indicates we make $softmax$ operation across the row direction.

$$M = \mathbf{A}\widetilde{X}, \qquad \widetilde{M} = tanh(w_f^c * M + b_f^c) \odot \sigma(w_g^c * M + b_g^c), \tag{1}$$

where $w_f^c \in \mathbb{R}^{1 \times m \times n}$, $w_g^c \in \mathbb{R}^{1 \times m \times n}$, $b_f^c \in \mathbb{R}^n$, and $b_g^c \in \mathbb{R}^n$ indicate the trainable parameters. '*' indicates the convolutional operation. '$\odot$' indicates element-wise product. Each row of the matrix $M$ represents a feature vector of a node, which is a weighted sum of the neighboring node features of the current node. $\widetilde{M} \in \mathbb{R}^{N \times n}$ indicates the output of GCN.

**The Computation of Conditional Centers.** Since $\widetilde{M}$ only captures relations between visual elements and does not have the capability to fully understand the image, we consider using NetVLAD [19, 4] to further enhance the representation of an image. By learning multiple centers, i.e., visual words, NetVLAD could use these centers to describe a scene [4]. However, these centers are learned based on the overall dataset and reflect the attributes of the dataset. In other words, these centers are independent of the current input data, which ignore the characteristic of the input data and reduce the accuracy of the representation. Here, we consider making an affine transformation for the initial centers and using these transformed centers to represent an image.

Concretely, we first define the initial centers $C = \{c_i \in \mathbb{R}^n, i = 1, ..., K\}$. Next, based on the current input query-image pairs, we make the affine transformation [34] for the initial centers.

$$\gamma = f(\langle \widetilde{M}, \widetilde{Y} \rangle), \qquad \beta = h(\langle \widetilde{M}, \widetilde{Y} \rangle), \qquad z_i = \gamma c_i + \beta, \tag{2}$$

where $\langle a, b \rangle$ represents the concatenation of $a$ and $b$. By stacking $Y$, we obtain $\widetilde{Y} \in \mathbb{R}^{N \times n}$. We separately use a two-layer convolutional network to define $f$ and $h$. $z_i \in \mathbb{R}^n$ indicates the $i$-th

generated conditional center. Here, we take the concatenated result of the representations of both input images and their corresponding queries as the input of $f$ and $h$ to compute parameter $\gamma$ and $\beta$. Since parameter $\gamma$ and $\beta$ are learned based on the input query-image pairs, these two parameters reflect the character of the current input data. Equipped with the affine transformation, the initial centers are made to move towards the input features, which improves the accuracy of the residual operation (Fig. 3(d)) of NetVLAD. As is shown in Fig. 3(b) and (c), after the affine transformation, the centers move towards the features (color circles). Finally, we use $Z = \{z_1, \cdots, z_K\}$ to indicate the new conditional centers.

**The Computation of GraphVLAD.** Next, we use $Z$ and $\widetilde{M}$ to perform NetVLAD operation,

$$D_j = \sum_{i=1}^{N} \frac{e^{w_j^T \widetilde{M}_i + b_j}}{\sum_{j'} e^{w_{j'}^T \widetilde{M}_i + b_{j'}}} (\widetilde{M}_i - z_j), \tag{3}$$

where $\{w_j\}$ and $\{b_j\}$ are sets of trainable parameters for each center $z_j$ and $j = 1, ..., K$. Finally, we use $D \in \mathbb{R}^{K \times n}$ to indicate the output of NetVLAD.

Besides, as is shown in Fig. 3(d), NetVLAD only captures relations between elements and centers. As NetVLAD is computed based on visual elements where relations are existed, we consider there should exist certain relations between outputs. Here, we employ GCN to capture these relations. Concretely, we first concatenate the NetVLAD output and conditional centers, i.e., $\widetilde{Z} = \langle z_1, \cdots, z_K \rangle$, $\widetilde{Z} \in \mathbb{R}^{K \times n}$, and $H = \langle D, \widetilde{Z} \rangle$. Then, we define an adjacency matrix for an undirected graph as $\mathbf{B} = softmax_r(HH^T) + I_d$. The following processes are the same as Eq. (1). Finally, we use $U \in \mathbb{R}^{K \times n}$ to indicate the output of GraphVLAD. By this operation, we obtain the global information of an image, which is as complementation of local object features $O \in \mathbb{R}^{L \times n}$ ($L$ indicates the number of objects) extracted by a pre-trained network and GCN network. Finally, the set $S = \{U, O\}$ is taken as the global representation of an image.

## 3.2 Contextualized Connectivity

The goal of contextualized connectivity is to not only capture the relevance between linguistic features and the global representation $S$, but also extract deep semantic existing in sentences according to visual information. Concretely, LSTM is employed to obtain representation $\widetilde{Q} \in \mathbb{R}^{P \times n}$ and $\widetilde{A} \in \mathbb{R}^{J \times n}$ of query and response, respectively, where $P$ and $J$ separately indicate the length of query and response. Next, we introduce the processing of the query. An attention operation is first used to obtain the relevance between the query and global representation.

$$F_{qu} = softmax_r(\widetilde{Q}U^T), \quad F_{qo} = softmax_r(\widetilde{Q}O^T), \quad Q_U = F_{qu}U, \quad Q_O = F_{qo}O. \tag{4}$$

Then, we take the concatenation of $Q_U$, $Q_O$, and $\widetilde{Q}$ as $Q_F \in \mathbb{R}^{P \times 3n}$. $U$ and $O$ are the output of the GraphVLAD. Here, we only obtain sequential features, rather than the structural information [41] which is helpful for a better understanding of the sentence semantic. Meanwhile, LSTM has the limitation of long-term information dilution [38], which weakens the capacity of the sentence representation. In this paper, we consider using GCN to extract structural information. Concretely, we define an adjacency matrix for an undirected graph as $\mathbf{Q} = softmax_r(Q_F Q_F^T) + I_d$. The following processes are the same as Eq. (1). Finally, we use $Q_g \in \mathbb{R}^{P \times n}$ to indicate the output of this network. The processing of responses is the same as that of queries. And the representation of response generated by a GCN network is defined as $A_g \in \mathbb{R}^{J \times n}$.

## 3.3 Directional Connectivity for Reasoning

Directional information is an important clue for cognitive reasoning. And using directional information could improve the accuracy of reasoning [32]. Here, we propose a semantic direction based GCN for reasoning. Concretely, we first use $\widetilde{A}$ to obtain the attention representation $Q_a \in \mathbb{R}^{J \times n}$ of $Q_g$. The processes are the same as Eq. (4). Then, $Q_a$ and $A_g$ are concatenated as $E_{qa} \in \mathbb{R}^{J \times 2n}$. Next, based on $E_{qa}$, we first try to learn the direction information.

$$D_{qa} = \phi(E_{qa}), \qquad G_t = D_{qa}D_{qa}^T, \qquad D_t = sign(G_t), \qquad V_e = softmax_r(abs(G_t)), \quad (5)$$

where $abs$ indicates the operation of absolute value. Here, $\phi$ is defined as a directional function, which is a one-layer convolutional network without activation. Besides, to learn the direction, we do not use ReLU activation at the last layer of the network $\phi$. By using the sign function, we obtain the direction $D_t$, where -1 and 1 separately indicate the negative and positive correlation. Next, based on the output $D_t$ of the sign function, we compute the adjacency matrix.

$$\mathbf{H} = D_t \odot V_e + I_d, \qquad M_t = \mathbf{H}E_{qa}, \qquad R_t = tanh(w_f^r * M_t + b_f^r) \odot \sigma(w_g^r * M_t + b_g^r), \quad (6)$$

where $\mathbf{H}$ indicates the adjacency matrix. $w_f^r \in \mathbb{R}^{1 \times 2n \times n}$, $w_g^r \in \mathbb{R}^{1 \times 2n \times n}$, $b_f^r \in \mathbb{R}^n$, and $b_g^r \in \mathbb{R}^n$ indicate the trainable parameters. Finally, we take $R_t \in \mathbb{R}^{J \times n}$ as the GCN output. By this operation, we could make our model not only learn the direction information between nodes, but also leverage the information in the computation of GCN, which results in accurate inference. In the experiment, compared with undirected GCN, our method could improve performance significantly.

### 3.4 Prediction Layer and Loss Function

After obtaining the output of the reasoning module, we concatenate $R_t$ and $\widetilde{A}$ across the channel dimension, i.e., $F_c = \langle R_t, \widetilde{A} \rangle$ and $F_c \in \mathbb{R}^{J \times 2n}$. Next, we compute a global vector representation $\widetilde{F} \in \mathbb{R}^{2n}$ via a max-pooling operation across the node dimension of $F_c$. This operation is helpful for getting a permutation invariant output and focusing on the impact of the graph structure [30]. Finally, we compute classification logits through a two-layer MLP with ReLU activation.

For VCR task, given a query-image pair, this task gives four response choices. In this paper, we train our model using a multi-class cross-entropy loss between the set of responses and the labels, i.e., $l(y, \hat{y}) = -\sum_{i=1}^{4} y_i log(\hat{y}_i)$, where $y$ denotes the ground truth and $\hat{y}$ is the predicted result.

## 4 Experiments

In this section, we evaluate our method on the VCR dataset. And this dataset contains 290k pairs of questions, answers, and rationales, over 110k unique movie scenes. Moreover, this task considers three modes, i.e., $Q \to A$ (given a question, select the correct answer), $QA \to R$ (given a question and the correct answer, select the correct rationale), and $Q \to AR$ (given a question, select the correct answer, then the correct rationale). For $Q \to AR$ mode, if it gets either the wrong answer or the wrong rationale, no points will be received. The code is available at https://github.com/AmingWu/CCN.

**Implementation details.** We use ResNet50 [15] to extract image and object features. BERT [11] is used as the word embedding. The feature map is $X \in \mathbb{R}^{12 \times 24 \times 512}$. The size of the hidden state of LSTM is set to 512. For Eq. (1), we use a one-layer GCN. And 32 centers are used to compute GraphVLAD. For Eq. (2), we separately use a two-layer network to define $f$ and $h$. Their parameters are all set to $1 \times 1024 \times 512$ and $1 \times 512 \times 512$. Next, we use a one-layer GCN to capture relations between centers. And the parameter settings of the GCN are the same as those of Eq. (1). For contextualized connectivity, we separately use a one-layer GCN to process query and response. Their parameter settings are the same as those of Eq. (1). For Eq. (5), a one-layer GCN is used for reasoning. Besides, the parameters of the network $\phi$ are set to $1 \times 1024 \times 512$. During training, we use Adam optimizer with a learning rate of $2 \times 10^{-3}$.

### 4.1 The Performance of Our Method

We evaluate our method on the three modes of VCR task. The results are shown in Table 1. We can see that some of state-of-the-art VQA methods, e.g., MUTAN [6] and BottomUpTopDown [1], do not perform well on this task. This shows that these VQA methods lack the inference ability, which results in unsatisfied performance on the task requiring high-level commonsense reasoning. Meanwhile, compared with the baseline method, on the three modes of VCR task, our method is 3.4%, 3.2%, and 4.4% higher than R2C, respectively. This shows that our method is effective.

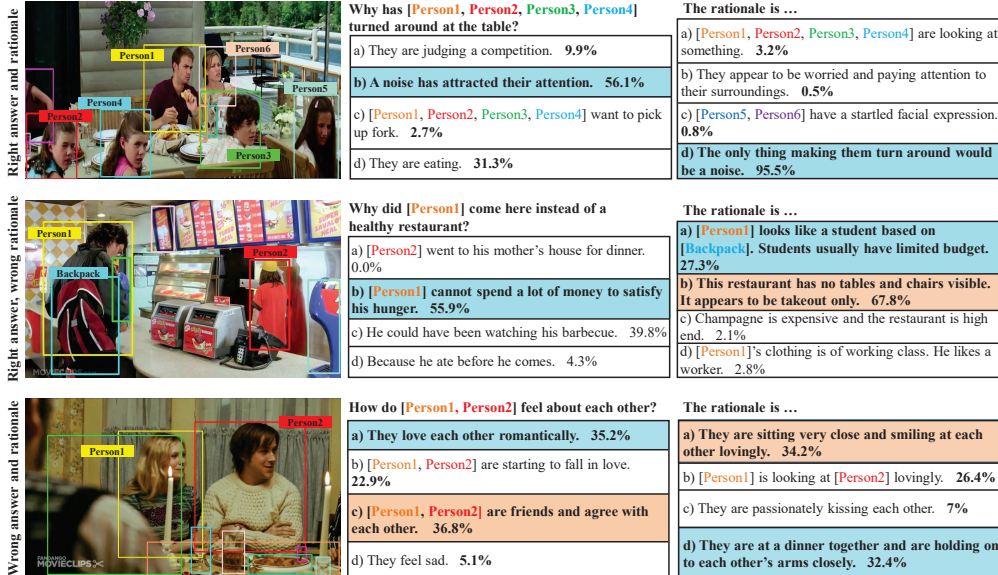

Figure 4: Qualitative examples from **CCN**. Correct choices are highlighted in blue. Incorrect inferences are in red. The number after each option indicates the score given by our model. The first row is a successful case. The second and last row correspond to two fail cases.

In Fig. 4, we show some qualitative examples. As is shown in these examples, compared with classical VQA dataset [3, 14], both questions and answers of VCR dataset are much more complex. Directly leveraging the recognition of visual content is difficult to choose the right answers and rationales. Besides, the first row of Fig. 4 is a successful case. Our model deduces the correct answer and its corresponding correct rationale with a high score. The second row shows a fail case, where the model chooses the right answer and the wrong rationale. However, the rationale chosen by our model is an explanation for the answer based on the understanding of the entire scene. Though from this view, the rationale is reasonable, compared with the ground truth, our rationale is slightly indirect and unclear. This shows when the visual reasoning involves more commonsense, the task of interpreting the answer is more difficult. Besides, though the model fails, the wrong rationale indeed matches the visual content, which shows the GraphVLAD module is helpful for obtaining an effective visual representation. The third row is also a fail case, where our model chooses the wrong answer and rationale. From these two fail cases, we can see that when the question, answer, and rationale involve much commonsense, the model is easy to make an error selection and indeed requires a strong ability of inference to choose the right answer and rationale. More examples can be found in Appendix.

Table 1: The performance of our CCN model on the VCR dataset.

| Model | $Q \rightarrow A$ | | $QA \rightarrow R$ | | $Q \rightarrow AR$ | |
|---|---|---|---|---|---|---|
| | Val | Test | Val | Test | Val | Test |
| Revisited VQA [18] | 39.4 | 40.5 | 34.0 | 33.7 | 13.5 | 13.8 |
| BottomUpTopDown [1] | 42.8 | 44.1 | 25.1 | 25.1 | 10.7 | 11.0 |
| MLB [20] | 45.5 | 46.2 | 36.1 | 36.8 | 17.0 | 17.2 |
| MUTAN [6] | 44.4 | 45.5 | 32.0 | 32.2 | 14.6 | 14.6 |
| R2C (baseline) [42] | 63.8 | 65.1 | 67.2 | 67.3 | 43.1 | 44.0 |
| **CCN** | **67.4** | **68.5** | **70.6** | **70.5** | **47.7** | **48.4** |

## 4.2 Ablation Analysis

In this section, based on the validation set, we make ablation analysis for our proposed conditional GraphVLAD, contextualized connectivity for extracting of the sentence semantic, and directional connectivity for reasoning, respectively.

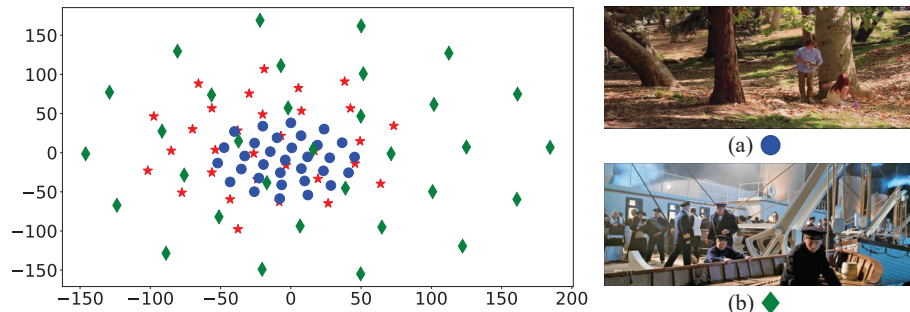

Figure 5: t-SNE plot of conditional centers. Here, the red pentagrams, blue circles, and green rhombuses indicate the initial centers and two different conditional centers, respectively. (a) and (b) are used to compute blue and green centers, respectively.

**GraphVLAD.** The number of centers is an important hyper-parameter for GraphVLAD. If few centers are used, it will weaken the representation ability of GraphVLAD. Conversely, if many centers are used, it will increase the number of parameters and computational costs. In $Q \rightarrow A$, $QA \rightarrow R$, and $Q \rightarrow AR$ modes, the performance of 16 centers and 48 centers separately is 66.4%, 69.2%, 46.4% and 67.1%, 69.8%, 46.9%. For our method, the performance of 32 centers is the best.

In Table 2, we analyze the effect of conditional centers and GCN for GraphVLAD. Here, 'No-C + No-G' indicates we use neither conditional centers nor GCN in the computation of GraphVLAD. And other components of our model are kept unchanged. We can see that employing conditional centers and GCN could improve performance significantly. Particularly, compared with NetVLAD corresponding to the case of 'No-C + No-G', our Conditional GraphVLAD outperforms NetVLAD significantly. This shows our method is effective. Besides, in Fig. 5, we show two t-SNE [25] examples of conditional centers. And the queries of Fig. 5(a) and (b) are "Who does the dog belong to?" and "What will happen after the person pushes the lifeboat over the edge of the ship?". We can see that the positions of centers vary depending on the visual content and its corresponding queries. When an image contains rich content and its corresponding query is complex, e.g., Fig. 5(b), in order to capture rich visual information to answer the query, these centers will learn to spread further apart from each other. Meanwhile, when the image content and its corresponding query contain relatively less information, e.g., Fig. 5(a), in order to focus on visual information which is related to the query, these centers will adaptively adjust to being more concentrated. In this way, we can obtain an effective visual representation, which is helpful for the following contextualization and reasoning.

**Contextualized Connectivity.** In this paper, we separately employ a GCN to capture the semantic of queries and responses. To prove this operation is effective, we compare it with a common operation, i.e., using a GCN to process the concatenation of vision, query, and response. In $Q \rightarrow A$, $QA \rightarrow R$, and $Q \rightarrow AR$ mode, the performance of the common operation is 66.5%, 68.1%, and 45.7%, which is obviously weaker than our method.

Table 2: Ablation analysis of GraphVLAD.

| Method | $Q \rightarrow A$ | $QA \rightarrow R$ | $Q \rightarrow AR$ |
|---|---|---|---|
| No-C + No-G | 65.8 | 68.3 | 45.6 |
| No-C | 66.5 | 69.6 | 46.6 |
| No-G | 66.9 | 69.4 | 46.5 |
| C + G | **67.4** | **70.6** | **47.7** |

Table 3: Ablation of Directional Reasoning.

| Method | $Q \rightarrow A$ | $QA \rightarrow R$ | $Q \rightarrow AR$ |
|---|---|---|---|
| No-R | 65.9 | 67.9 | 45.3 |
| LSTM-R | 64.8 | 67.1 | 43.9 |
| GCN | 66.5 | 69.4 | 46.4 |
| D-GCN | **67.4** | **70.6** | **47.7** |

**Directional Connectivity for Reasoning.** In this paper, we propose a directional reasoning method. We compare our method with other reasoning methods. The results are shown in Table 3. Here, 'No-R' indicates we do not use reasoning. 'LSTM-R', 'GCN', and 'D-GCN' indicate reasoning based on LSTM, undirected GCN, and directed GCN, respectively. And other components of our model are kept the same. We can see that for the method without reasoning, the performance is obviously weak. This shows reasoning is a necessary step for our method. Besides, the performance of the reasoning based on LSTM is also weak. This shows that LSTM could not capture complex relations effectively.

Finally, compared with undirected GCN reasoning, our directional GCN outperforms it significantly. This shows using directional information in reasoning could improve the accuracy of inference.

## 5 Conclution

We propose a cognition connectivity network for directional visual commonsense reasoning. This model mainly includes visual neuron connectivity, contextualized connectivity, and directional connectivity for reasoning. Particularly, for visual neuron connectivity, we propose a conditional GraphVLAD module to represent an image. Meanwhile, we propose a directional GCN for reasoning. The experimental results demonstrate the effectiveness of our method. Particularly, in the $Q \to AR$ mode, our method is 4.4% higher than R2C.

## Acknowledgement

This work is supported by the NSFC (under Grant 61876130, 61932009, U1509206).

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
