[Supplementary Material]

# Connective Cognition Network for Directional Visual Commonsense Reasoning: Supplementary Material

Aming Wu[1][*]    Linchao Zhu[2]    Yahong Han[1][†]    Yi Yang[2]

[1]College of Intelligence and Computing, Tianjin University, Tianjin, China
[2]ReLER, University of Technology Sydney, Australia
{tjwam,yahong}@tju.edu.cn, {Linchao.Zhu, yi.yang}@uts.edu.au

**t-SNE Plots of Conditional Centers.** In Fig. 1, we show two examples of conditional centers. Compared with the initial centers, the distribution of the conditional centers of Fig. 1(a) is more diverse, while that of the conditional centers of Fig. 1(b) is less diverse. This shows that when the image content is rich and its corresponding query is complex, to better represent an image, the distribution of the conditional centers will be much more diverse than that of the initial centers. Meanwhile, when the image content and its corresponding query contain relatively little information, the distribution of the conditional centers will be less diverse to focus on the image information which is related to the query. This further proves the conditional centers are meaningful.

Figure 1: t-SNE plots of conditional centers. The red pentagrams indicate the initial centers. And the blue circles indicate the conditional centers. For these two plots, we all use the default t-SNE settings.

**Examples from the CCN.** In Fig. 2, 3, and 4, we show many examples. And we can see that the queries, answers, and rationales are complex. Our CCN method not only gives the correct answer but also deduces the correct rationale with a high score. This shows that our method is effective.

---

[*]This work was done when Aming Wu visited ReLER Lab, UTS.
[†]Corresponding author

**What [Person3] do next?**

| |
|---|
| a) **[Person3] pays for the items he wants.  54.7%** |
| b) He walks out the building.    **29.1%** |
| c) Put his hands up slowly to show he is not reaching for a weapon.    **13.2%** |
| d) After finishing speaking to [Person2] moves to the chair behind him, and sinks into the chair while running his hand over his face in disbelief.    **3.0%** |

**The rationale is …**

| |
|---|
| a) **[Person1] is a cashier which is a person you give money to in exchange for products.    66.1%** |
| b) [Person1] is breaking into a mine where such items as valuable metals or buried treasure would be located.  **1.5%** |
| c) [Person2] wants to purchase new boots but does not know the proper size.    **22.9%** |
| d) [Person1] is wearing a uniform and standing behind a register.    **9.5%** |

**Why are [Person1, Person2] and [Person3] have brunch?**

| |
|---|
| a) [Person1, Person2] and [Person3] are just enjoying the view while out for a walk.    **10.9%** |
| b) [Person1, Person2] and [Person3] are employees of [Person1].    **0.4%** |
| c) They enjoy reading books and consider reading to be a worthwhile hobby.    **27.3%** |
| d) **They are on vacation at a resort.    61.4%** |

**The rationale is …**

| |
|---|
| a) There is sand everywhere and the ocean is in the backdrop. Many of the men have their shirts off and there are women wearing bikinis.    **5.2%** |
| b) They have just married and it is custom to enjoy a vacation after a wedding.    **23.7%** |
| c) **They are dressed in clothing someone might wear on vacation. There is a lake behind them and a large building can be seen in the background.    41.9%** |
| d) It is light food with a lot of fruits. It seems like in the afternoon which is brunch time.    **29.2%** |

Figure 2: Qualitative examples from **CCN**. The number after each option indicates the score given by our model. The correct results are highlighted in blue.

**What will [Person1, Person3] do if [Person2] catches up to them?**

| |
|---|
| a) [Person1, Person3] will start to pick up their paces and run faster if [Person2] catches up.  **97.5%** |
| b) [Person1, Person3] will fly away.  **1.2%** |
| c) [Person1, Person3] will scream for [Person2].  **1.0%** |
| d) [Person1, Person3] hug, and follow [Person2] to their destination.  **0.3%** |

**The rationale is …**

| |
|---|
| a) If [Person2] closes the distance between himself and [Person1, Person3], then [Person1, Person3] will be concerned that [Person2] is going to push ahead of them, so they will run faster because they want to keep in the lead.  **99.3%** |
| b) [Person2] looks like he is really picking up his legs to try to set himself apart from the other runners.  **0.2%** |
| c) [Person1, Person3] flank [Person2] as he walks between them.  **0.4%** |
| d) If [Person2] gets short, [Person1, Person3] will have a chance of catching him on foot.  **0.1%** |

**What is [Person1] going to do next?**

| |
|---|
| a) [Person7] is going to shoot a gun.  **0.0%** |
| b) [Person9] is going to jump into the water.  **0.0%** |
| c) [Person1] is going to hit [Chair].  **100.0%** |
| d) [Person7] is going to try to calm [Person1] down.  **0.0%** |

**The rationale is …**

| |
|---|
| a) [Person1] is moving his arms back in a swinging motion and aiming towards [Chair].  **79.0%** |
| b) [Person7] is always reeling from the punch, a chair shot can cause a lot of pain and knock people out. [Person1] is expressing rage on his face.  **0.1%** |
| c) If [Person1] keeps walking, the chair in his path will trip them.  **0.3%** |
| d) [Person1] is holding [Chair] with his left hand, which would make sense if he stole it from [Person9], who is also on his left. [Person1] is trying to get away from [Person9] and has his right fist clenched as if he is getting ready to throw a punch.  **20.6%** |

Figure 3: Qualitative examples from **CCN**. The number after each option indicates the score given by our model. The correct results are highlighted in blue.

**Why does [Person1] have laptop in front on him?**

| |
|---|
| **a) [Person1] is using the laptop to record everything that [Person2] is saying, so they can review it later.    100.0%** |
| b) [Person2] is helping to monitor the race and is keeping track of its participants.    **0.0%** |
| c) [Person2] is sitting in a control room .    **0.0%** |
| d) [Person2] is playing a virtual reality game.    **0.0%** |

**The rationale is …**

| |
|---|
| a) Meeting minutes reflect the actions taken during a business or organizational meeting. Minutes are typically recorded by a secretary and become an essential part of the records.    **0.4%** |
| **b) Instead of writing the information [Person2] is giving them, [Person1] is saving it on the laptop.    98.6%** |
| c) Often data from instruments is important to record to prove that someone has accomplished something. The data can be used by others to try to recreate an experiment for example.    **0.4%** |
| d) [Person5, Person7] are making sure that what [Person2] is saying is good for them as well as [Person1].    **0.6%** |

**What are [Person1, Person2, Person3] all doing at the table together?**

| |
|---|
| a) They are sitting down to a meal.    **0.4%** |
| b) They are playing cards.    **0.2%** |
| **c) They are all having breakfast together.    99.4%** |
| d) They are at the dinner table together because they are each other dates for the party.    **0.0%** |

**The rationale is …**

| |
|---|
| a) [Bottle, Cup] are filled with orange juice, a common breakfast time drink.    **28.8%** |
| b) There are multiple empty bottles of maple syrup on [Chair] in front of [Person1, Person3], and maple syrup is generally only has in notable quantities for breakfast.    **1.8%** |
| c) We know that [Person1, Person2, Person3] are having a meal together, because there is food on the table.    **10.7%** |
| **d) [Person1, Person2, Person3] are all seated around the table with fruits, juice, and coffee in front of each of them. There is sun but not bright sun.    58.7%** |

Figure 4: Qualitative examples from **CCN**. The number after each option indicates the score given by our model. The correct results are highlighted in blue.