[Reviews · NeurIPS 2019]

Reviewer 1



Originality: The paper proposes a novel model for the recently introduced VCR task. The main novelty of the proposed model lies in the component GraphVLAD and directional GCN modules. The paper describes that one of the closest works to this work is that of Narsimhan et al., NeurIPS 2018 that used GCN to infer answers in VQA, however that work constructs an undirected graph, ignoring the directional information between the graph nodes. This paper uses directed graph instead and shows the usefulness of incorporating directional information. It would be good for this paper to include more related work on GraphVLAD front. Quality: The paper evaluates the proposed approach on the VCR dataset and compares with the baselines and previous state-of-the-art, demonstrating how the proposed work improves the previous best performance significantly. The paper also reports ablation studies ablation each individual contributed module of the proposed CCN model, demonstrating the usefulness of each component. It would be useful if the paper could throw some light on the failure modes of the proposed model. Clarity: I would like the authors to clarify the following -- 1. Based on the description in Sec 3.1, Graph VLAD needs the query LSTM representation as an input. However, this is not consistent with Fig. 2. Can authors please rectify the Fig. 2? 2. In Eq. 3, it seems like it should be b_j’, instead of b_k’ in the denominator. Can authors please comment on this and rectify accordingly? 3. Sec 3.1 – how are conditional centers (z1, …., zK) initialized? 4. L149 – can authors provide further clarification on how object features are extracted. What kind of pre-trained network is used and how is that different from the network used for extracting the feature map X for the image which is used in GraphVLAD? 5. How is the query representation Y define in L100 different from that Q~ defined in L154? Significance: The proposed model is novel and interesting for a novel and useful task. The idea of GraphVLAD module and directional reasoning seem to be impactful and could be used for other vision and language tasks as well. The experiments demonstrate that the proposed model improves the state-of-the-art on VCR significantly. --- Post-rebuttal comments ---- The authors have responded to and addressed most of my clarification questions. It would have been nice to see some related work on GraphVLAD front in the rebuttal too (very briefly) but I am trusting the authors to do justice to this in the final version of the paper. Regarding the concerns from my fellow reviewers -- 1. Connections to brain -- I don't feel too strongly about this. 2. Results on VQA-CP / VQA -- I treat VCR as a different task from VQA / VQA-CP as it focusses more on commonsense reasoning which is not there in VQA / VQA-CP enough. It would be nice to test the proposed model on VQA / VQA-CP as well (which authors have done (only on VQA-CP), however their VQA-CP results are not beating the state-of-the-art (47.70 on VQA-CP v2 by Selvaraju et al., ICCV 2019 and 41.17 on VQA-CP v2 by Ramakrishnan et al., NeurIPS 2018)); however I do not consider lack of beating state-of-art on VQA / VQA-CP to be a reason to reject this paper. 3. Glove in language model -- the proposed model uses BERT and beats the previous state-of-art using BERT (R2C). Zellers et al. already show that VQA models using Glove perform much worse than VQA models using BERT on VCR. So given that it has been established that BERT is much better than Glove for VCR, I am not sure why it is not enough to just keep using and comparing with BERT and not show results on Glove (assuming fair comparison with previous state-of-art).

Reviewer 2



The paper proposes a new model for VQA, and explores it in the context of the VCR dataset for visual commonsense reasoning. The model uses two main components: first NetVLAD [1] and then Graph Convolution Network [2] in order to propagate contextual information across the visual objects, which is interesting and different from prior work in VQA (especially the use of NetVLAD). Experimental results are good and The paper is also generally well-written and structured clearly (except some part of the model description as discussed below). Model - The paper motivates the new model mainly by comparing it to the neuronal connectivity within the brain. I feel that in this case the comparison is not very justified/convincing. While certainly relational reasoning is important for tasks of visual question answering, I would be happy if more evidence could be presented and discussed in the paper to establish the proposed connection between the relational model in the paper and the operation of the brain. - NetVLAD: Throughout the model description section, the discussion assumes familiarity of the readers with VLAD and NetVLAD - I think it could be helpful to discuss them at least briefly either in the Related Work section or in the beginning of the subsection about “The Computation of Conditional Centers” (Page 4): e.g. to explain in high-level what the method does and provide a bit more detail on how the new model uses it. Experiments - The paper provides experiments only over a new dataset for commonsense reasoning called VCR [3]. It could thus be really useful if experiments would be provided also for more standard datasets, such as VQA1/2 or the more recent VQA-CP, to allow better comparison to many existing approaches for visual question answering. - Most VQA models so far tend to use word embedding + LSTM to represent language - Could you please perform an ablation experiment for your model using such approach? While it’s very likely that BERT helps improve performance, and indeed baselines for VCR use BERT, I believe it is important to also have results based on the more currently common approach to again allow better comparison to most existing VQA approaches. - (Page 8) The t-SNE visualization is explained kind of briefly and it is not totally clear to me what is the main insight/conclusion that could be derived from that: we see that for one image (blue) the centers are more concentrated and for the other (green) they are spread further apart from each other - it would be good if the authors could discuss in more detail why such a thing might happen or what aspects of the image impact the density of the centers, or how the density affects the downstream reasoning performance. Clarity - (page 2) The second paragraph that gives an overview of the model is quite long and hard to follow. I think it would be good to make this part more structured: maybe splitting the paragraph into multiple ones that present each of the stages the model goes through. [1] Arandjelovic, Relja, Petr Gronat, Akihiko Torii, Tomas Pajdla, and Josef Sivic. "NetVLAD: CNN architecture for weakly supervised place recognition." In Proceedings of the IEEE conference on computer vision and pattern recognition, pp. 5297-5307. 2016. [2] Kipf, Thomas N., and Max Welling. "Semi-supervised classification with graph convolutional networks." arXiv preprint arXiv:1609.02907 (2016).

Reviewer 3



This paper proposes neural network models to represent stepwise cognitive processes similar to humans' for visual understanding to deal with VCR dataset. The idea of their methods is very interesting and the results are good enough to outperform other state-of-the-art methods. However, this paper would be better if the following issues are improved. Firstly, the use of notation is rather confusing and not consistent. It would be nice to write it more concise and consistent. Secondly, as reported in [38, Zellers et al., CVPR'19], GloVe is used as language models for the comparative methods in Table 1. Since there is no comment about them, it may mislead that all of the other models adopt BERT same to the proposed models. It would be better to explain explicitly and report the performance of both cases. Thirdly, there is a lack of explanation in some experiment settings. How is 'Rationale' configured in each mode? For example, in case of QA->R, will A be concatenated with Q in textual form after finishing Q->A? If yes, the latter characters can be discarded depending on P and J values, what values are used for them? Optionally, it will be helpful to report the generalized performance by showing the results on well-known Visual QA datasets such as VQA 1.0 and 2.0. Here is minor comment: In reference, the order of authors in [32] is not correct with the original one. Also, some commas are missing between the author names. In eq.3, k' in the denominator is j'.

[Author Response · NeurIPS 2019]

We thank the reviewers for their recognition of our work and helpful comments.

**Q1/Reviewer#1:** "More related work on GraphVLAD front". **A:** We will modify our paper to cite more related work.

**Q2/Reviewer#1:** "Throw some light on the failure modes". **A:** In Fig. 4, we have shown two failure examples and
given some discussions (Line 220-227). Taking the 2nd row as an example, when the inference of correct rationale
requires a strong ability of reasoning with the commonsense, the model might fail. The insights are: Firstly, when the
visual reasoning involves more commonsense, the task of interpreting the answer is more difficult. Secondly, though the
model fails, the wrong rationale indeed matches the visual content, which shows GraphVLAD module is helpful for
obtaining an effective visual representation.

**Q3/Reviewer#1:** "On the modification of Fig. 2". **A:** Thanks for your advice. We will rectify Fig. 2.

**Q4/Reviewer#1:** "On the denominator of Eq. 3". **A:** Thank you. In the denominator, it should be $b_{j'}$, instead of $b_{k'}$.

**Q5/Reviewer#1:** "On the initialization of centers". **A:** We use random uniform initialization on the interval $[0, 1)$.

**Q6/Reviewer#1:** "On the extraction of object features and feature map". **A:** The visual feature of each object is
Roi-Aligned from its bounding region [2]. And we use ResNet50 as the backbone in Mask R-CNN [2]. By performing
a Max-pooling operation on the output of the third block of ResNet50, we obtain $X$ to compute GraphVLAD.

**Q7/Reviewer#1:** "On the difference between the representation $Y$ and $\widetilde{Q}$". **A:** In this paper, $Y$ is a feature vector which
indicates the output of LSTM at the last time step. $\widetilde{Q}$ is an output matrix which contains the output of LSTM at each
time step. And the first dimension of $\widetilde{Q}$ indicates the length of the query.

**Q1/Reviewer#2:** "The connection between our model and the brain". **A:** In Line 37, recent studies [28] on brain
networks have suggested that brain function or cognition can be described as the global and dynamic integration of
local neuronal connectivity. Since we used GCN in each module, our CCN could be regarded as a hierarchical GCN.
The bottom layer is used to capture local relations. As the layer increases, our model could use dynamic connections to
integrate local relations. Finally, the top layer performs reasoning based on the global integration of all the relations.
This process is similar to that of the brain cognition.

**Q2/Reviewer#2:** "On the explanation of NetVLAD". **A:** We will explain more details of NetVLAD in our paper.

**Q3/Reviewer#2:** "The performance of VQA-CPv2". **A:** We test our CCN on the VQA-CPv2 dataset [1]. We do not
change the hyper-parameters tuned on VCR. The accuracy of our method is 39.44%, which outperforms the baseline
method [1] by 8%. Recently, [32] specially designed a multimodal fusion module for VQA. It used a pairwise modeling
component to further update the multimodal representation with multiple iterations. We obtained comparable results to
[32] (39.54%) in VQA-CPv2. This shows our method could be readily applied to standard VQA task.

**Q4/Reviewer#2:** "The result of the method based on word embedding and LSTM". **A:** We conducted the baseline that
used "word embedding + LSTM". The LSTM encoded the query and response. In $Q \rightarrow A$, $QA \rightarrow R$, and $Q \rightarrow AR$
mode, the performance on the validation set is 57.3%, 60.1%, and 36.5%. Our CCN significantly outperforms it.

**Q5/Reviewer#2:** "More interpretations about the t-SNE visualization". **A:** In this paper, we want to use conditional
centers to capture the characteristics of the current input data. When an image contains rich content and its corresponding
query is complex, e.g., Fig. 5(b), in order to capture rich visual information to answer the query, these centers will
learn to spread further apart from each other. Meanwhile, when the image content and its corresponding query contain
relatively less information, e.g., Fig. 5(a), in order to focus on visual information which is related to the query, these
centers will adaptively adjust to being more concentrated. In this way, we can obtain an effective visual representation,
which is helpful for the following contextualization and reasoning.

**Q1/Reviewer#3:** "On notations". **A:** We will modify our paper to make notations much more concise and consistent.

**Q2/Reviewer#3:** "On the use of BERT". **A:** We will modify our paper to explain the details of the other models clearly.
Besides, we tried some VQA models, e.g., MUTAN [5] and MLB [18], and used BERT as word embedding. However,
we found R2C model (baseline in our paper) outperformed them by around 3% in terms of $Q \rightarrow AR$ performance.

**Q3/Reviewer#3:** "On the explanation in some experimental settings". **A:** In Line 193-196, we have given the setting
details of each mode. For the case of $QA \rightarrow R$, it uses the question and **corresponding ground-truth answer** to
predict rationale. And the answer is concatenated with the question in textual form. In this case, $P$ indicates the length
of the concatenation of the question and the answer. And $J$ indicates the length of the rationale.

**Q4/Reviewer#3:** "On the generalized performance". **A:** Here, we test our method on the VQA-CPv2 dataset [1] which
is the improved version of VQA 2.0. The accuracy of our method is 39.44%, which outperforms the baseline method
[1] by 8%. This shows our method could be readily applied to standard VQA task. Please refer to our response to
Q3/Reviewer#2 for details.

## References

[1] A. Agrawal, D. Batra, D. Parikh, and A. Kembhavi. Don't just assume; look and answer: Overcoming priors for visual question answering. In *CVPR*, pages
4971–4980, 2018.
[2] K. He, G. Gkioxari, P. Dollár, and R. Girshick. Mask r-cnn. In *ICCV*, pages 2961–2969, 2017.


[Meta-Review · NeurIPS 2019]

After considering the author response and discussing the submission, all reviewers voted to accept. Most reviewers shared a concern that the neurological connection is relatively weak and encourage authors to discuss it in looser terms like "inspired by". Reviewers generally found the paper's claims well established.